# RELATIONAL CONVOLUTIONAL NETWORKS: A FRAMEWORK FOR LEARNING REPRESENTATIONS OF HIERARCHICAL RELATIONS

## ABSTRACT

A maturing area of research in deep learning is the development of architectures that can learn explicit representations of relational features. In this paper, we focus on the problem of learning representations of *hierarchical* relations, proposing an architectural framework we call "relational convolutional networks". Given a sequence of objects, a "multi-dimensional inner product relation" module produces a relation tensor describing all pairwise relations. A "relational convolution" layer then transforms the relation tensor into a sequence of new objects, each describing the relations within some group of objects at the previous layer. Graphlet filters, analogous to filters in convolutional neural networks, represent a template of relations against which the relation tensor is compared at each grouping. Repeating this yields representations of higher-order, hierarchical relations. We present the motivation and details of the architecture, together with a set of experiments to demonstrate how relational convolutional networks can provide an effective framework for modeling relational tasks that have hierarchical structure.

## 1 INTRODUCTION

Modeling of relations between objects is important to a range of machine learning problems. For instance, an image analysis application might rely on comparing objects in terms of relations that extract color, size, or texture features; a natural language task may use relations that are based on syntactic or semantic features of pairs of words. To enable efficient learning of relational information, it is important to explore learning architectures that support processing of relations in a natural, expressive, and efficient manner.

Compositionality, the ability to compose modules together to build iteratively more complex feature maps, is key to the success of deep representation learning. For example, in a feed forward network, each layer builds on the one before, and in a CNN, each convolution builds an iteratively more complex feature map (Zeiler & Fergus, 2014). So far, work on relational representation learning has been limited to "flat" first-order architectures. In this work, we propose a compositional framework for learning hierarchical relational representations, which we call "relational convolutional networks."

A schematic of the proposed architecture to support hierarchical relational learning is shown in Figure 1. The key idea involves formalizing a notion of "convolving" a relation tensor, describing the pairwise relations in a sequence of objects, with a "graphlet filter" which represents a template of relations between subsets of objects. Each composition of those operations computes relational features of a higher order.

In a series of experiments, we show how relational convolutional neural networks provide an effective framework (and inductive bias) for relational learning. We first carry out experiments on "relational games" proposed as a benchmark for relational reasoning by (Shanahan et al., 2020). This consists of a suite of binary classification tasks for identifying abstract relational rules between a set of objects represented as images. We next carry out experiments on a version of the SET game, which requires processing of higher-order relations between multiple attributes on a set of cards. For both tasks, relational convolutional networks are able to achieve more sample efficient learning compared to Transformers, as well as other architectures that have been specifically developed for relational learning.

## 1.1 RELATED WORK

Several previous works have considered the design of machine learning architectures which support the representation of relational information, in various forms (Battaglia et al., 2018; Palm et al., 2018; Zhang et al., 2019).

**Graph Neural Networks.** Graph neural networks (GNNs) are a class of neural network architectures which operate on graphs and process "relational" data (Kipf et al., 2018; Kipf & Welling, 2017; Niepert et al., 2016; Schlichtkrull et al., 2017; Veličković et al., 2017). A defining feature of the GNN model is its use of a form of neural message-passing, wherein the hidden representation of a node is updated as a function of the hidden representations of its neighbors (Gilmer et al., 2017). Typical examples of tasks which GNNs are applied to include node classification, graph classification, and link prediction (Hamilton, 2020).

In GNNs, the 'relations' are given to the model via edges in a graph. In contrast, our architecture, as well as the explicitly relational architectures described below, operate on collections of objects without any relations given as input. Instead, such relational architectures must infer the relevant relations from the objects themselves. Still, graph neural networks can be applied to relational tasks by passing in the collection of

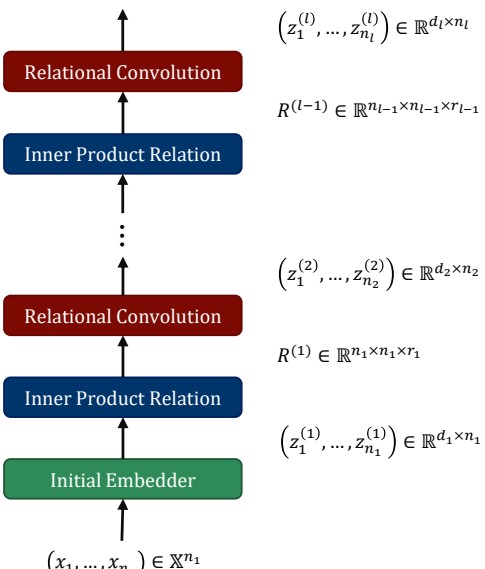

Figure 1: Proposed architecture for relational convolutional networks. Hierarchical relations are modeled by iteratively computing pairwise relations between objects and convolving the resultant relation tensor with graphlet filters representing templates of relations between subsets of objects.

objects along with a complete graph. A Transformer Encoder can be thought of as a special case of this architecture, and hence is the representative baseline we compare against in our experiments.

**Attention as modeling relations** Several works have proposed architectures with the ability to model relations by incorporating an attention mechanism (Locatello et al., 2020; Santoro et al., 2018; Vaswani et al., 2017; Veličković et al., 2017; Zambaldi et al., 2018). Attention mechanisms model relations between objects implicitly as an intermediate step in a form of neural message-passing in order to update the representation of each object as a function of its context.

**Emerging literature on "explicitly relational" architectures.** There exists a growing literature on neural architectures which aim to explicitly model relational information between objects. An early example is (Santoro et al., 2017). Shanahan et al. (2020) proposes the PrediNet architecture, which aims to learn relational representations which are compatible with predicate logic. Webb et al. (2021) proposes ESBN, a recurrent neural network augmented with external memory with a memory-write operation which factors representations into 'sensory' and 'relational'. Kerg et al. (2022) proposes CoRelNet, a simple architecture based on 'similarity scores' which aims to distill the relational inductive biases discovered in previous work into a minimal architecture.

We aim to contribute to this literature by proposing a framework for learning hierarchical relational representations in a natural, interpretable, sample-efficient, and parameter-efficient manner.

## 2 MULTI-DIMENSIONAL INNER PRODUCT RELATION MODULE

A relation function is a function which maps a pair of objects $x, y \in \mathcal{X}$ to a vector representing the relation between the two objects. For example, a relation may represent the information "$x$ has the same color as $y$", "$x$ is larger than $y$", and "$x$ is to the left of $y$". In principle, this can be modeled by an arbitrary learnable function on the concatenation of the two objects' representations. For example, (Santoro et al., 2017) models relations by MLPs applied to the concatenation of pairs of

objects, $g_\theta(x, y)$. While this approach may work in some cases, it is missing some crucial inductive biases. In particular, there is no constraint that the learned pairwise function is in fact *relational*— $g_\theta(x, y)$ may just as well represent non-relational information like "$x$ is bright" and "$y$ is small", as opposed to relational information like "$x$ is larger than $y$".

Recent work has explored using *inner products* to model relations between objects (Altabaa et al., 2023; Kerg et al., 2022; Webb et al., 2021). The advantage of such an approach is that it provides added pressure to learn explicitly relational representations, disentangling relational information from attributes of individual objects. In particular, it induces a geometry on the object space $\mathcal{X}$ which allows objects to be described in relation to each other. For example, in the symmetric case, the inner product $\langle \varphi(x), \varphi(y) \rangle$ induces a metric on $\mathcal{X}$. In fact, the relation $\langle \varphi(x), \varphi(y) \rangle$ attaches well-defined notions of distance, angles, and orthogonality to the space $\mathcal{X}$.

More generally, we can allow for multi-dimensional relations by having multiple encoding functions, each extracting a feature to compute a relation on. Furthermore, we can allow for asymmetric relations by having different encoding functions for each object. Hence, we model relations by,

$$r(x, y) = (\langle \varphi_1(x), \psi_1(y) \rangle, \ldots, \langle \varphi_{d_r}(x), \psi_{d_r}(y) \rangle) \in \mathbb{R}^{d_r}, \tag{1}$$

where $\varphi_1, \psi_1, \ldots, \varphi_{d_r}, \psi_{d_r}$ are learnable functions. For each dimension of the relation function, the maps $\varphi_k, \psi_k$ extract a particular attribute of the objects which is then compared by the inner product.

To promote weight sharing, we can have one common non-linear map $\varphi$ across all dimensions along with different linear maps for each object and each dimension of the relation. That is,

$$r(x, y) = \left( \left\langle W_1^{(1)} \varphi(x), W_2^{(1)} \varphi(y) \right\rangle, \ldots, \left\langle W_1^{(d_r)} \varphi(x), W_2^{(d_r)} \varphi(y) \right\rangle \right), \tag{2}$$

where the learnable parameters are $\varphi$ and $W_1^{(k)}, W_2^{(k)}, k \in [d_r]$. $\varphi : \mathcal{X} \to \mathbb{R}^{d_\varphi}$ may be an MLP, for example, and $W_1^{(k)}, W_2^{(k)}$ are $d_{\text{proj}} \times d_\varphi$ matrices. The class of functions realizable by Equation (2) is the same as Equation (1) but enables greater weight sharing.

The "Multi-dimensional Inner Product Relation" (MD-IPR) module receives a sequence of objects $x_1, \ldots, x_m$ as input and models the pairwise relations between them by Equation (2), returning an $m \times m \times d_r$ relation tensor, $R[i, j] = r(x_i, x_j)$, describing the relations between each pair of objects.

## 3 RELATIONAL CONVOLUTIONS WITH GRAPHLET FILTERS

### 3.1 RELATIONAL CONVOLUTIONS WITH DISCRETE GROUPS

Suppose that we have a sequence of objects $(x_1, \ldots, x_m) \in (\mathbb{R}^d)^m$ and a relation tensor $R \in \mathbb{R}^{m \times m \times d_r}$ describing the pairwise relations between them (obtained by a MD-IPR layer). The relational convolution operation we will define does two things: 1) extracts features of the relations between groups of objects using pairwise relations 2) transforms the relation tensor back into a sequence of objects, allowing it be composed with another relational layer to compute higher-order relations.

Fix some filter size $s < n$, where $s$ is a hyperparameter of the relational convolution layer. One 'filter' is given by the *graphlet* $f_1 \in \mathbb{R}^{s \times s \times d_r}$. This is a 'template' for the pairwise relations between $s$ objects. Note that the dimension of the relations in this filter matches that of the input relation tensor. Let $g \subset [n]$ be a subset of the objects of size $s$. Suppose for now that $g$ is an ordered set (i.e., the group $(1, 2, 3)$ is different from the group $(2, 3, 1)$). Then, denote the relation sub-tensor given by this (ordered) subset by $R[g] := [R[i, j]]_{i,j \in g}$. We define the 'relational inner product' between this relation subtensor and the filter $f_1$ by

$$\langle R[g], f_1 \rangle_{\text{rel}} := \sum_{i,j \in g} \langle R[i, j], f_1[i, j] \rangle_{\mathbb{R}^{d_r}} = \sum_{i,j \in g} \sum_{k \in [d_r]} R[i, j, k] f_1[i, j, k]. \tag{3}$$

This is simply the inner product in the corresponding euclidean space $\mathbb{R}^{s^2 d_r}$. This quantity represents how much the relations within the objects in $g$ match the relations in the template $f_1$.

The relational convolution layer has $n_f$ filters (a hyperparameter). Denote the collection of filters by $\boldsymbol{f} = \left( f_1, \ldots, f_{n_f} \right) \in \mathbb{R}^{s \times s \times d_r \times n_f}$, which we call a *graphlet filter*. We define the relational inner

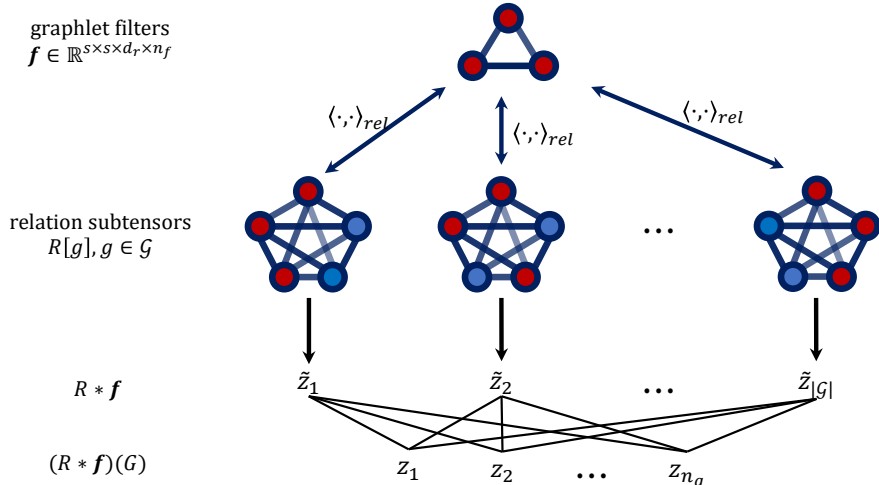

graphlet filters
$\boldsymbol{f} \in \mathbb{R}^{s \times s \times d_r \times n_f}$

relation subtensors
$R[g], g \in \mathcal{G}$

$R * \boldsymbol{f}$

$(R * \boldsymbol{f})(G)$

Figure 2: A depiction of the relational convolution operation. The input is a relation tensor $R \in \mathbb{R}^{m \times m \times d_r}$. A learned set of graphlet filters $\boldsymbol{f} \in \mathbb{R}^{s \times s \times d_r \times n_f}$ are compared to each relation subtensor $R[g], g \in \mathcal{G}$, producing the relational convolution $R * \boldsymbol{f}$. With a group matrix $G \in \mathbb{R}^{m \times n_g}$, $(R * \boldsymbol{f})(G)$ represents the relational information in a set of $n_g$ "soft groups".

product of a relation subtensor $R[g]$ with a graphlet filter $\boldsymbol{f}$ as the $n_f$-dimensional vector consisting of the relational inner products with each individual filter,

$$\langle R[g], \boldsymbol{f} \rangle_{\mathrm{rel}} := \begin{pmatrix} \langle R[g], f_1 \rangle_{\mathrm{rel}} \\ \vdots \\ \langle R[g], f_{n_f} \rangle_{\mathrm{rel}} \end{pmatrix} \in \mathbb{R}^{n_f}. \tag{4}$$

This vector summarizes various aspects of the relations within a group, captured by several different filters[1].Each filter corresponds to one dimension in the final relation-summarizing vector for the group $g$. This is reminiscent of convolutional neural networks, where each filter gives us one channel in the output tensor.

We can also define a symmetric variant of the relational inner product which is invariant to the ordering of the elements in $g$. This can be done by pooling over all permutations of $g$. In particular, we suggest max-pooling and average-pooling, although any set-aggregator would be valid. We denote the permutation-invariant relational inner product by $\langle R[g], f_1 \rangle_{\mathrm{rel,sym}}$,

$$\langle R[g], \boldsymbol{f} \rangle_{\mathrm{rel,sym}} := \mathrm{Pool}\left(\left\{ \langle R[g'], \boldsymbol{f} \rangle_{\mathrm{rel}} : g' \in g! \right\}\right), \tag{5}$$

where $g!$ denotes the set of permutations of the group $g$. Recall that each $\langle R[g'], \boldsymbol{f} \rangle_{\mathrm{rel}}$ is $n_f$-dimensional, and the pooling is done independently for each dimension.

For a given group $g \subset [m]$, the relational inner product with a graphlet filter, $\langle R[g], \boldsymbol{f} \rangle_{\mathrm{rel}}$, gives us a vector summarizing the relational patterns inside that group. We aim to get a sequence of objects which each describes the relational patterns within each group of interest. Let $\mathcal{G}$ be a set of size-$s$ groups of the the $m$ objects. The relational convolution between a relation tensor $R$ and a relational graphlet filter $\boldsymbol{f}$ is the sequence of relational inner products with each group in $\mathcal{G}$,

$$R * \boldsymbol{f} := \left( \langle R[g], \boldsymbol{f} \rangle_{\mathrm{rel}} \right)_{g \in \mathcal{G}} \equiv \left( z_1, ..., z_{|\mathcal{G}|} \right) \in \left( \mathbb{R}^{n_f} \right)^{|\mathcal{G}|} \tag{6}$$

$\mathcal{G}$ is a pre-specified hyperparameter of the relational convolution operation. The choice depends on the usecase. If some prior information is known about reasonable groupings, this can be encoded in $\mathcal{G}$. When $m$ is small and no prior information is available, a reasonable choice might be the the set of all combinations of size $s$. When $m$ is large, considering all combinations will be intractable. One solution is to consider a random sample of combinations. In the next subsections, we consider the problem of *learning* the relevant groups.

---

[1]We have overloaded notation, but will use the convention that a collection of filters is denoted by a bold symbol to distinguish between the two forms of the relational inner product.

## 3.2 RELATIONAL CONVOLUTIONS WITH 'SOFT' GROUPS

In the above formulation, the groups are 'discrete'. Having discrete groups can be desirable for interpretability, if the relevant groupings are known a priori or if considering every possible grouping is computationally and statistically feasible. However, if the relevant groupings are not known, then considering all possible combinations results in a rapid growth of the number of objects at each layer.

In order to address these issues, we propose *explicitly modeling groups*. This allows us to control the number of objects in the output sequence of a relational convolution operation such that only relevant groups are considered. In the next section, we outline some ways to model 'soft groups' using *grouping layers*. These layers take a sequence of objects and/or the relation tensor as input and produce a 'group matrix' $G \in \mathbb{R}^{m \times n_g}$ representing $n_g$ 'soft groups'. The $(i, j)$-th entry of the group matrix represents the degree to which the $i$-th object belongs to the $j$-th group. The number of groups $n_g$ is a configurable hyperparameter of the grouping layers. For the remainder of this subsection, we assume that the group matrix $G$ is given as input to the relational convolution layer.

Consider the group matrix $G \in \mathbb{R}^{m \times n_g}$ and filters $\boldsymbol{f}$ of size $s$. First, we use $G$ to compute a "group-match score" for each discrete group $g$ of size $s$ (e.g., $g \in \mathcal{G} = \binom{[m]}{s}$). This is done via

$$G \leftarrow \text{SoftPlus}(G)$$

$$\alpha_{gk} \leftarrow \text{Normalize}\left(\left[\prod_{i \in g} G[i, k]\right]_{g \in \mathcal{G}}\right), \quad g \in \mathcal{G}, k \in [n_g], \tag{7}$$

where the soft-plus function is $\text{Softplus}(x) = \log(\exp(x + 1))$, applied elementwise. This has the effect of making the group matrix $G$ non-negative which is needed for the product of its elements to represent a "group-match score". The product inside the softmax is over elements in the discrete group $g \in \mathcal{G}$. Hence, it will be large whenever the soft group $G_k := G[:, k]$ aligns with the discrete group $g$. $\text{Normalize}(\cdot)$ normalizes the group match scores so that $\sum_g \alpha_{gk} = 1$. We propose the use of sparse normalizers (Laha et al., 2018) so that only a sparse subset of discrete groups in $\mathcal{G}$ contribute to each soft group (see also Appendix B.2). In our experiments, we use 'sparsemax' (Martins & Astudillo, 2016). Thus, $\alpha_{gk}$ is a normalized "group-match score" indicating the degree to which the discrete group $g$ matches the given soft group $G_k$.

Now, we can define the 'soft' relational inner product *given* the soft group $G_k$ by

$$\langle R, \boldsymbol{f} \,|\, G_k \rangle_R := \sum_{g \in \mathcal{G}} \alpha_{gk} \langle R[g], \boldsymbol{f} \rangle_{\text{rel}}. \tag{8}$$

This notation should be read as "the relational inner product of the relation tensor $R$ with the graphlet filters $\boldsymbol{f}$ given the group $G_k$". This expression is essentially a convex combination of the relational inner product with all possible discrete groups weighted by how much they match the soft group $G_k$.

With this modification, the number of objects in the output sequence is fixed and controlled by the number of groups, $n_g$ (which is a hyperparameter). The output sequence of the relational convolution given groups $G$ is now given by

$$(R * \boldsymbol{f})(G) = \left(\langle R, \boldsymbol{f} \,|\, G_1 \rangle_{\text{rel}}, \ldots, \langle R, \boldsymbol{f} \,|\, G_{n_g} \rangle_{\text{rel}}\right) \in (\mathbb{R}^{n_f})^{n_g}. \tag{9}$$

## 4 GROUPING LAYERS

A grouping layer is a layer which outputs a group matrix $G \in \mathbb{R}^{m \times n_g}$ representing the degree to which each object $i \in [m]$ belongs to each group $j \in [n_g]$. The number of groups $n_g$ is a configurable hyperparameter. We briefly describe some proposals for grouping layers with different properties.

**Temporal (Positional) Grouping.** In the temporal grouping layer, the groups are a function only of the temporal order of the objects. This can be achieved by learning the group matrix $G$ directly as a parameter of the model. $G$ will be optimized along with the rest of the model parameters as a function of its effect on the relational convolution layer. Temporal grouping would be appropriate in situations where the order in which objects appear is predetermined and indicates the relevant groups. For example, objects that are positionally close to each other may be grouped together.

**Feature-based Grouping.** In a feature-based grouping layer, the group(s) to which each object belongs is a function of that object's features (and position). That is,

$$G \leftarrow \begin{bmatrix} \varphi(1, x_1)^\top \\ \vdots \\ \varphi(m, x_m)^\top \end{bmatrix} \in \mathbb{R}^{m \times n_g}, \tag{10}$$

where $\varphi : [m] \times \mathcal{X} \to \mathbb{R}^{n_g}$ is a learnable function which maps an object's temporal order $i$ and feature representation $x_i$ to a $n_g$-dimensional group membership vector where the $j$th entry of the vector represents the degree to which the object belongs to the $j$th group. For example $\varphi$ can be a multi-layer perceptron of the form $\varphi(i, x) = \mathrm{MLP}(\mathrm{concat}(e_i, x))$. Feature-based grouping may be useful in situations where group membership can be determined for each object using only that object's features, irrespective of the context of the other objects in the sequence.

**Context-aware Grouping.** In some applications, the group(s) to which each object belongs to may depend on the full context of the other objects in the sequence. One way to model this is to use a message-passing neural network to update the representations of each object, incorporating the context of the other objects in the sequence. Then, a multi-layer perceptron is applied to each encoded object to produce the group membership vector for that object.

$$E_i \leftarrow \mathrm{MessagePassing}\left(x_i, \{x_1, \ldots, x_m\}\right), \ i \in [m]$$
$$G \leftarrow \begin{bmatrix} \mathrm{MLP}(E_1)^\top \\ \vdots \\ \mathrm{MLP}(E_m)^\top \end{bmatrix} \in \mathbb{R}^{m \times n_g}. \tag{11}$$

This is the most general form of grouping, as it encompasses the previous two forms as special cases. The updated representation of each object $E_i$ now contains any relevant information about the other objects which should be considered in computing its group membership vector. One simple and effective option for the message-passing operation is to use self-attention. In this case, since an inner product relation layer precedes the relational convolution layer, the relation tensor can be re-used in the self-attention operation to compute attention scores.

## 5 EXPERIMENTS

### 5.1 RELATIONAL GAMES

The *relational games* dataset was contributed as a benchmark for relational reasoning by (Shanahan et al., 2020). It consists of a family of binary classification tasks for identifying abstract relational rules between a set of objects represented as images. The objects consist of three sets of simple geometric shapes, referred to as "pentominoes", "hexominoes", and "stripes". The objects are arranged in a $3 \times 3$ grid. Each task corresponds to some relationship between the objects (see Figure 3), and the target is to classify whether the relationship holds among a given sequence of objects or not.

In our experiments, we evaluate out-of-distribution generalization by training all models on the pentominoes objects and evaluating on the hexominoes and stripes objects. The input to all models is presented as a sequence of 9 objects, each represented as a $12 \times 12 \times 3$ RGB image. In all models, the objects are processed independently by a CNN with a shared architecture. The individually-processed sequence of objects is then passed to the relation-processing component of the model. The results are then flattened and passed through an MLP with a shared architecture to produce the final prediction. We compare four models: a relational convolutional network (abbreviated RelConvNet), CoRelNet (Kerg et al., 2022), PrediNet (Shanahan et al., 2020), and a Transformer (Vaswani et al., 2017). Further, we evaluate variants of RelConvNet with learned grouping layers (temporal, feature-based, and contextual). The architectural details of each model are described in Appendix A.

The pentominoes split is used for training, and the hexominoes and stripes splits are used to test out-of-distribution generalization after training. We train for 50 epochs using the categorical cross-entropy loss and the Adam optimizer with learning rate $0.001$, $\beta_1 = 0.9, \beta_2 = 0.999, \epsilon = 10^{-7}$. We use a batch size of 512. For each model and task, we run 5 trials with different random seeds.

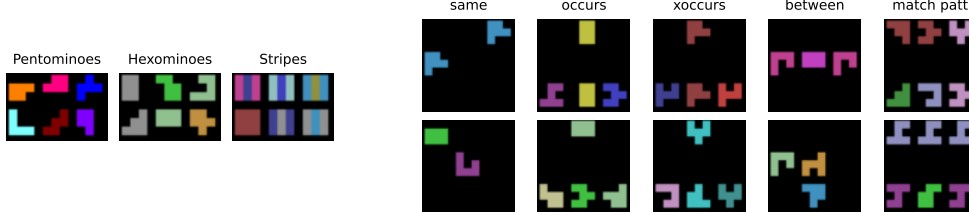

Figure 3: Relational games dataset. **Left** Examples of objects from each split. **Right** Examples of problem instances for each task. The first row is an example where the relation holds and the second row is an example where the relation does not hold.

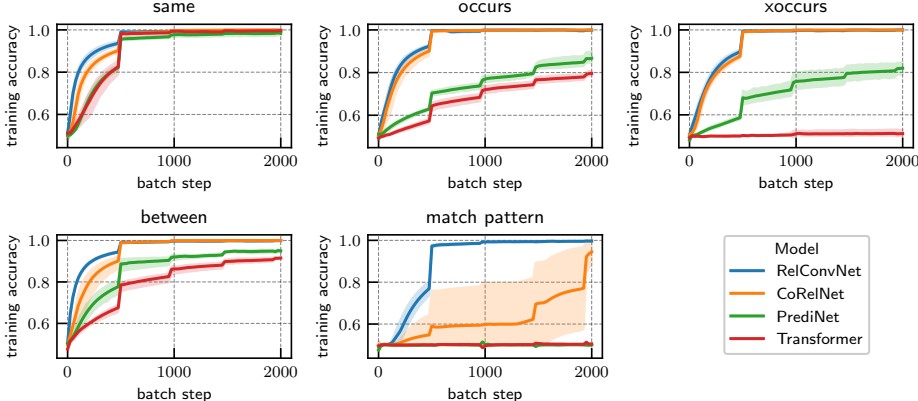

Figure 4: Training curves, up to 2,000 batch steps, for each relational games task. Solid lines indicate the mean over 5 trials and the shaded regions indicate a bootstrap 95% confidence interval.

**Sample efficiency.** We observe that the relational inductive biases of RelConvNet, and relational models more generally, grant a significant advantage in sample-efficiency. Figure 4 shows the training accuracy over the first 2,000 batches for each model. RelConvNet, CoRelNet, and PrediNet are explicitly relational architectures, whereas the Transformer is not. The Transformer is able to process relational information through its attention mechanism, but this information is entangled with the features of individual objects (which, for these relational tasks, is extraneous information). The Transformer consistently requires the largest amount of data to learn the relational games tasks. PrediNet tends to be more sample-efficient. RelConvNet and CoRelNet are the most sample-efficient, with RelConvNet only slightly more sample-efficient on most tasks.

On the 'match pattern' task, however, RelConvNet is significantly more sample-efficient. We attribute this to the fact that RelConvNet is able to model higher-order relations through its relational convolution module. The 'match pattern' task can be thought of as a simple second-order relational task—it involves computing relations between two groups of objects, and comparing the relations within the two groups. The relational convolution module naturally models this kind of situation since it learns representations of the relations among subsets of objects.

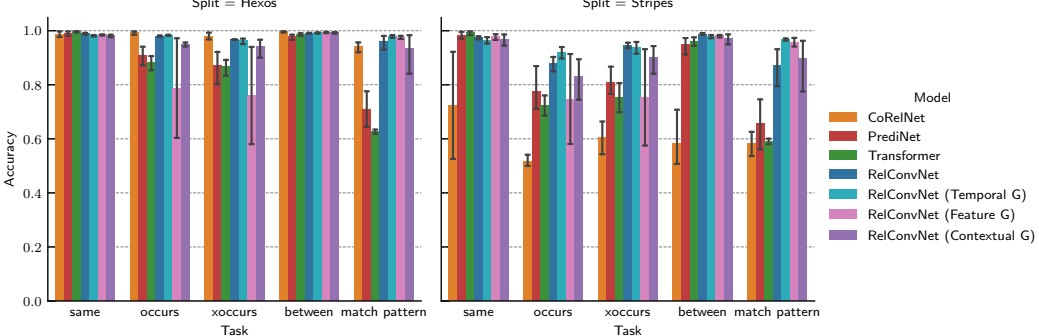

Figure 5: Out-of-distribution generalization on hold-out object sets. Bar heights indicate the mean over 5 trials and the error bars indicate a bootstrap 95% confidence interval.

**Out-of-distribution generalization.** Figure 5 reports model performance on the two hold-out object sets after training. On the hexominoes objects, which are similar-looking to the pentominoes objects used for training, RelConvNet and CoRelNet do nearly perfectly. PrediNet and the Transformer do well on the simpler tasks, but struggle with the more difficult 'match pattern' task. The stripes objects are visually more distinct from the training split objects, making generalization more difficult. We observe an overall drop in performance for all models. The drop is particularly dramatic for CoRelNet[2]. We conjecture that this is due to CoRelNet's inability to model multi-dimensional relations, necessitating that all relational information is squeezed into a scalar quantity. The separation between RelConvNet and the other models is largest on the "match pattern" task of the stripes split (the most difficult task and the most difficult generalization split). Here, RelConvNet (with temporal grouping) maintains a mean accuracy of 97% while the other models drop below 65%. We attribute this to RelConvNet's ability to naturally represent higher-order relations and model groupings of objects. This split is also where the addition of learned grouping layers yield improvements over the variant of RelConvNet with only discrete groups. The grouping layers allow the model to represent the latent groups which exist in the task, enabling improved out-of-distribution generalization.

## 5.2 'SET': GROUPING AND COMPOSITIONALITY IN RELATIONAL REASONING

'SET' is a card game which forms a simple but challenging relational task. The 'objects' are a set cards each representing four attributes which can take one of three values each. 'Color' can be red, green, or purple; 'number' can be one, two, or three; 'shape' can be diamond, squiggle, or oval; and 'fill' can be solid, striped, or empty. A 'set' is a triplet of cards such that each attribute is either the same on all three cards or different on all three cards. Figure 6a shows a sample of SET cards.

In SET, the task is: given a hand of $k > 3$ cards, find a 'set' among them (typically, SET is played with $k = 12$, with two players competing to find a 'set' first). This task is deceptively challenging, and is representative of the type of relational reasoning that humans excel at but machine learning systems still struggle with. To solve the task, one must process the sensory information of individual cards to identify the values of each attribute, then somehow search over combinations of cards and reason about the relations between them. Importantly, this type of relational reasoning requires attending over several attributes and relations simultaneously while representing some notion of 'groups'. The construct of relational convolutions proposed in this paper is a step towards developing machine learning systems which can perform this kind of relational reasoning.

In this section, we evaluate RelConvNet on a task based on 'SET' and compare it to several baselines. The task is: given a sequence of $k = 5$ images of SET cards, determine whether or not they contain a 'set'. All models share the common architecture $(x_1, \ldots, x_k) \to \texttt{CNN} \to \{\cdot\} \to \texttt{MLP} \to \hat{y}$, where $\{\cdot\}$ is RelConvNet, CoRelNet, PrediNet, or a Transformer. The architectures are identical to the previous section with the exception that RelConvNet uses the permutation-invariant version of the relational inner product (eq. (5)) with max-pooling, and the projection dimension is changed to 16 to better match the larger embedding dimension of the card images. The CNN embedder is pre-trained on the task of classifying the four attributes of the cards and an intermediate layer is used to generate embeddings of dimension 64 for each card. The output MLP architecture is shared across all models, and consists of two hidden layers with 64 and 32 neurons, respectively, and ReLU activations.

In SET, there exists $\binom{81}{3} = 85\,320$ triplets of cards, of which $1\,080$ are a 'set'. We partition the 'sets' into training (70%), validation (15%), and test (15%) sets. The training, validation, and test datasets are generated by sampling $k$-tuples of cards such that with probability $1/2$ the $k$-tuple does not contain a set, and with probability $1/2$ it contains a set among the corresponding partition of sets. Partitioning the data in this way allows us to measure the models' ability to "learn the rule" and identify new unseen 'sets'. We train for 100 epochs with the same loss, optimizer, and batch size as the experiments in the previous section. Figure 6b shows the hold-out test accuracy for each model. Figures 6c and 6d show the training and validation accuracy over the course of training.

We observe that RelConvNet is able to learn the task and generalize to new 'sets' with near-perfect accuracy. On the other hand, CoRelNet and the Transformer have accuracies only slightly better than random guessing on the test set, while PrediNet learns nothing that generalizes to the test set. The

---

[2]The experiments in (Kerg et al., 2022) on the relational games benchmark use a technique called "context normalization" (Webb et al., 2020) as a preprocessing step. We choose not to use this technique since it is an added confounder. We discuss this more in Appendix E.

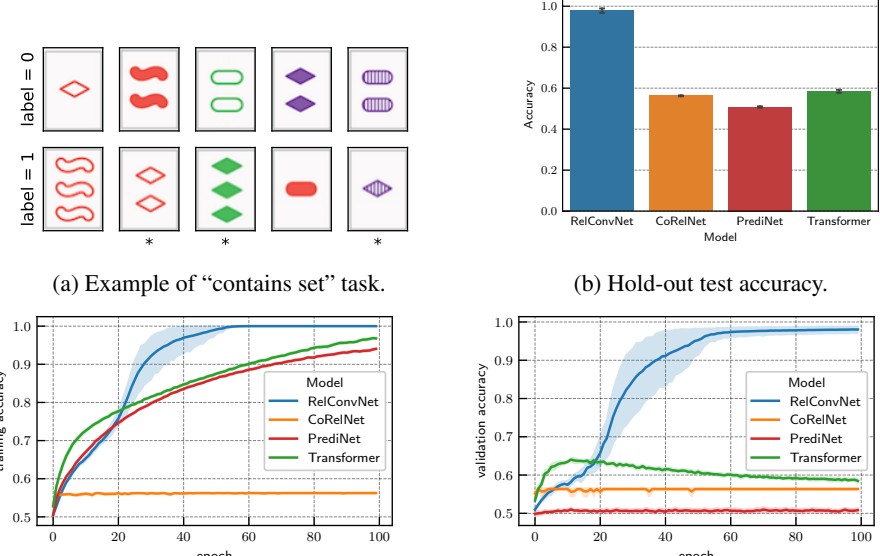

(a) Example of "contains set" task.

(b) Hold-out test accuracy.

(c) Training accuracy over the course of training. (d) Validation accuracy over the course of training.

Figure 6: Results of "contains set" experiments. Bar height/solid lines indicate the mean over 10 trials and error bars/shaded regions indicate 95% bootstrap confidence intervals.

picture becomes more clear when looking at training curves. While the Transformer and PrediNet are eventually able to fit the training data, they are unable to "learn the rule" in a way that generalizes to the validation or test sets. This suggests that the Transformer architecture has the general-purpose function-approximation capabilities to fit a wide array of sequence-tasks, but it does not have the right inductive biases for specialized relational reasoning. A possible explanation is that relations are processed only implicitly through its attention mechanism, rather than explicitly via a relation tensor. Moreover, it does not naturally support reasoning about the relations between groups of objects. These are strengths of the relational convolutional networks architecture.

## 6 DISCUSSION

**Summary.** In this paper, we proposed a framework for hierarchical relational representation learning via a novel relational convolution operation. The relational convolution operation we propose here is a 'convolution' in the sense that it considers a patch of the relation tensor, given by a group, and compares the relations within it to a template graphlet filter via an appropriately-defined inner product. This is analogous to convolutional neural networks, where we would compare an image filter against different patches of the input image. Since the same graphlet filters are used for all groupings, the relational convolution operation implements a form of *parameter-sharing* which yields improved sample-efficiency and generalization.

An important feature of the relational convolution operation is its *interpretability*. The filters $f = (f_1, \ldots, f_{n_f})$ are each a particular pattern of relations between $s$ objects. Each object in the output of a relational convolution $R * f$ represents the degree to which the relations in the group $g$ match the patterns in each filter. By iteratively applying inner product relation layers and relational convolution layers, we obtain an architecture which naturally models *hierarchical* relations.

**Limitations and future work.** The tasks considered here are solvable by modeling only second-order relations at most. We observe that the relational convolutional networks architecture saturates the relational games benchmark of (Shanahan et al., 2020). While the "contains set" task demonstrates a sharp separation between relational convolutional networks and existing baselines, this task too only involves second-order relations, and does not fully test the abilities of the framework. A more thorough evaluation of this architecture, and future architectures for modeling hierarchical relations,

would require the development of new benchmark tasks and datasets which involve a larger number of objects and higher-order relations. This is a non-trivial task that we leave for future work.

The experiments considered here are synthetic relational tasks designed for a controlled evaluation. In more realistic settings, we envision relational convolutional networks as modules embedded in a broader architecture. For example, the relational convolutions framework can be naturally integrated into a graph neural network by using the relational convolution outputs as the node inputs and the relation tensor as the graph inputs. Similarly, the framework can be integrated into a Transformer-based model for general sequence modeling tasks by having the decoder attend to the sequence of relational objects produced by relational convolutions.

CODE AND DATA

Code and experimental logs are available at: `https://[PlaceHolder]`

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

## A    EXPERIMENTS SUPPLEMENT

### A.1    RELATIONAL GAMES

Table 1 contains text descriptions of each task in the relational games dataset in the experiments of Section 5.1. Table 2 contains a description of the architectures of each model (or shared component) in the experiments. Table 3 contains a description of the hyperparameters of the grouping layers corresponding to the 'Temporal G', 'Feature G', and 'Contextual G' RelConvNet models. All grouping layers have 16 soft groups, and the normalizer in Equation (7) is sparsemax. Table 4 reports the accuracy on the hold-out object sets (i.e., the numbers depicted in Figure 5 of the main text).

| Task | Description |
|------|-------------|
| `same` | Two random cells out of nine are occupied by an object. They are the "same" if they have the same color, shape, and orientation (i.e., identical image) |
| `occurs` | The top row contains one object and the bottom row contains three objects. The "occurs" relationship holds if at least one of the objects in the bottom row is the same as the object in the top row. |
| `xoccurs` | Same as occurs, but the relationship holds if exactly one of the objects in the bottom row is the same as the object in the top row. |
| `between` | The grid is occupied by three objects in a line (horizontal or vertical). The "between" relationship holds if the outer objects are the same. |
| `row match pattern` | The first and third rows of the grid are occupied by three objects each. The "match pattern" relationship holds if the relation pattern in each row is the same (e.g., AAA, AAB, ABC, etc.) |

Table 1: Relational games tasks.

| Model / Component | Architecture |
|-------------------|--------------|
| Common CNN Embedder | `Conv2D → MaxPool2D → Conv2D → MaxPool2D → Flatten`. `Conv2D`: num filters = 16, filter size = $3 \times 3$, activation = relu. `MaxPool2D`: stride = 2. |
| RelConvNet | `CNN → MD-IPR → RelConv → Flatten → MLP`. `MD-IPR`: relation dim = 16, projection dim = 4, symmetric. `RelConv`: num filters = 16, filter size = 3, discrete groups = combinations. |
| CoRelNet | `CNN → CoRelNet → Flatten → MLP`. Standard CoRelNet has no hyperparameters. |
| PrediNet | `CNN → PrediNet → Flatten → MLP`. `PrediNet`: key dim = 4, number of heads = 4, num relations = 16. |
| Transformer | `CNN → TransformerEncoder → AveragePooling → MLP`. `TransformerEncoder`: num layers = 1, num heads = 8, feedforward intermediate size = 32, activation = relu. |
| Common output MLP | `Dense(64, 'relu') → Dense(2)`. |

Table 2: Model architectures for relational games experiments.

| Grouping layer | Description |
|----------------|-------------|
| Temporal Grouping | $G$ learned directly as parameter. |
| Feature-based Grouping | $G$ produced by MLP with two hidden layers of size 32 and relu activation. |
| Contextual Grouping | $G$ produced by message-passing (via Transformer Encoder Block) followed by an MLP (eq. (11)). Encoder Block has 4 heads, inner dim = 32, and relu activation. MLP same as feature-based grouping. |

Table 3: hyperparameters of grouping layers.

| Task | Model | Hexos Accuracy | Stripes Accuracy |
|---|---|---|---|
| same | CoRelNet | $0.988 \pm 0.006$ | $0.724 \pm 0.112$ |
| | PrediNet | $0.990 \pm 0.004$ | $0.983 \pm 0.007$ |
| | Transformer | $0.997 \pm 0.001$ | $0.993 \pm 0.004$ |
| | RelConvNet | $0.989 \pm 0.002$ | $0.974 \pm 0.003$ |
| | RelConvNet (Temporal G) | $0.981 \pm 0.002$ | $0.965 \pm 0.007$ |
| | RelConvNet (Feature G) | $0.985 \pm 0.001$ | $0.978 \pm 0.006$ |
| | RelConvNet (Contextual G) | $0.981 \pm 0.002$ | $0.969 \pm 0.012$ |
| occurs | CoRelNet | $0.992 \pm 0.004$ | $0.518 \pm 0.012$ |
| | PrediNet | $0.907 \pm 0.020$ | $0.775 \pm 0.046$ |
| | Transformer | $0.881 \pm 0.015$ | $0.724 \pm 0.021$ |
| | RelConvNet | $0.980 \pm 0.001$ | $0.880 \pm 0.015$ |
| | RelConvNet (Temporal G) | $0.983 \pm 0.001$ | $0.920 \pm 0.012$ |
| | RelConvNet (Feature G) | $0.788 \pm 0.112$ | $0.747 \pm 0.099$ |
| | RelConvNet (Contextual G) | $0.951 \pm 0.004$ | $0.830 \pm 0.044$ |
| xoccurs | CoRelNet | $0.980 \pm 0.007$ | $0.606 \pm 0.035$ |
| | PrediNet | $0.872 \pm 0.036$ | $0.810 \pm 0.028$ |
| | Transformer | $0.867 \pm 0.017$ | $0.753 \pm 0.031$ |
| | RelConvNet | $0.967 \pm 0.001$ | $0.946 \pm 0.006$ |
| | RelConvNet (Temporal G) | $0.963 \pm 0.006$ | $0.939 \pm 0.012$ |
| | RelConvNet (Feature G) | $0.760 \pm 0.107$ | $0.754 \pm 0.103$ |
| | RelConvNet (Contextual G) | $0.942 \pm 0.020$ | $0.901 \pm 0.029$ |
| between | CoRelNet | $0.995 \pm 0.001$ | $0.582 \pm 0.063$ |
| | PrediNet | $0.978 \pm 0.006$ | $0.950 \pm 0.019$ |
| | Transformer | $0.986 \pm 0.003$ | $0.961 \pm 0.010$ |
| | RelConvNet | $0.991 \pm 0.001$ | $0.988 \pm 0.002$ |
| | RelConvNet (Temporal G) | $0.992 \pm 0.001$ | $0.979 \pm 0.004$ |
| | RelConvNet (Feature G) | $0.993 \pm 0.001$ | $0.981 \pm 0.003$ |
| | RelConvNet (Contextual G) | $0.992 \pm 0.002$ | $0.972 \pm 0.012$ |
| match pattern | CoRelNet | $0.942 \pm 0.011$ | $0.581 \pm 0.026$ |
| | PrediNet | $0.710 \pm 0.040$ | $0.658 \pm 0.053$ |
| | Transformer | $0.627 \pm 0.005$ | $0.591 \pm 0.006$ |
| | RelConvNet | $0.961 \pm 0.015$ | $0.870 \pm 0.041$ |
| | RelConvNet (Temporal G) | $0.979 \pm 0.003$ | $0.967 \pm 0.002$ |
| | RelConvNet (Feature G) | $0.976 \pm 0.003$ | $0.958 \pm 0.009$ |
| | RelConvNet (Contextual G) | $0.935 \pm 0.047$ | $0.896 \pm 0.060$ |

Table 4: Out-of-distribution generalization results on relational games. We report means $\pm$ standard error of mean over 5 trials. These are the numbers associated with Figure 5.

## A.2 SET

Table 6 reports the test accuracies in the "contains set" task of Section 5.2 (i.e., the numbers depicted in Figure 6b of the main text).

## B COMPUTATIONAL EFFICIENCY AND PARAMETER EFFICIENCY

### B.1 PARAMETER EFFICIENCY

The parameter count of both MD-IPR and relational convolution layers does not scale with the number of input objects or the number of groups considered. Instead, the parameters are shared across objects and groups. This has computational and statistical benefits. In particular, it reduces memory usage and the number of weight updates during backpropagation, as well as making the statistical estimation of a good choice of parameters easier.

| Model / Component | Architecture |
|---|---|
| Common CNN Embedder | `Conv2D → MaxPool2D → Conv2D → MaxPool2D → Flatten →` `Dense(64, 'relu') → Dense(64, 'tanh')`. `Conv2D`: num filters = 32, filter size = $5 \times 5$, activation = relu. `MaxPool2D`: stride = 4. |
| RelConvNet | `CNN → MD-IPR → RelConv → Flatten → MLP`. `MD-IPR`: relation dim = 16, projection dim = 16, symmetric. `RelConv`: num filters = 16, filter size = 3, discrete groups = combinations, symmetric relational inner product with 'max' aggregator. |
| CoRelNet | `CNN → CoRelNet → Flatten → MLP`. Standard CoRelNet has no hyperparameters. |
| PrediNet | `CNN → PrediNet → Flatten → MLP`. `PrediNet`: key dim = 4, number of heads = 4, num relations = 16. |
| Transformer | `CNN → TransformerEncoder → AveragePooling → MLP`. `TransformerEncoder`: num layers = 1, num heads = 8, feedforward intermediate size = 32, activation = relu. |
| Common output MLP | `Dense(64, 'relu') → Dense(32, 'relu') → Dense(2)`. |

Table 5: Model architectures for "contains set" experiments.

| Model | Accuracy |
|---|---|
| CoRelNet | $0.563 \pm 0.001$ |
| PrediNet | $0.508 \pm 0.002$ |
| RelConvNet | $0.979 \pm 0.006$ |
| Transformer | $0.584 \pm 0.004$ |

Table 6: Hold-out test accuracy on "contains set" task. We report means $\pm$ standard error of mean over 10 trials. These are the numbers associated with Figure 6b.

**MD-IPR layer.** In a multi-dimensional inner product relation module, the parameters are the left and right projection maps, $W_1^{(k)}, W_2^{(k)} \in \mathbb{R}^{d_{\mathrm{proj}} \times d_{\mathrm{in}}}, k = 1, \ldots, d_r$, where the projection dimension $d_{\mathrm{proj}}$ and the relation dimension $d_r$ are hyperparameters, and $d_{\mathrm{in}}$ is the dimensionality of the input objects ( Equation (2)). Hence, the parameter count of a MD-IPR layer is $d_r \times d_{\mathrm{proj}} \times d_{\mathrm{in}}$. Observe that while this scales with the dimensionality of the input objects $d_{\mathrm{in}}$, it does not scale with the number of objects $m$—the same encoders are shared across all pairs of objects.

**Relational Convolution layer.** The parameters in a relational convolution layer are the graphlet filters $\boldsymbol{f} \in \mathbb{R}^{s \times s \times d_r \times n_f}$. Observe that the parameter count is independent of the number of groups $|\mathcal{G}|$ or even the number of objects $m$. The same graphlet filters are applied to the relation subtensor corresponding to each grouping of objects. This is similar to how convolutional neural networks apply the same filters to all patches of an image. The parameter count of a relational convolution layer depends only on the filter size and number of filters, which are hyperparameters.

## B.2 COMPUTATIONAL CONSIDERATIONS

The MD-IPR module computes an $m \times m$ relation tensor, where $m$ is the number of input objects. This quadratic computational and memory cost can be avoided under assumptions of sparsity in the discrete groups $\mathcal{G}$ or group assignment matrix $G$. Such optimizations will be useful in problems involving a large number of objects.

Consider first the case relational convolution with discrete groups. The output of the relational convolution layer is $R * \boldsymbol{f} = (\langle R[g], \boldsymbol{f} \rangle_{\mathrm{rel}})_{g \in \mathcal{G}}$. For each $g \in \mathcal{G}$, $R[g]$ encodes the relations between objects inside of $g$. Hence, rather than computing the full $m \times m$ relation tensor, we can instead compute only,

$$\mathcal{R} := \{R_{ij} \colon \exists\, g \in \mathcal{G} \text{ such that } i, j \in g\}.$$

When $\mathcal{G}$ is sparse and structured, $\mathcal{R}$ is sparse.

In the case of relational convolution with (learned) soft groups, the same can be achieved via a sparse selection in the group match score $\alpha_{gk}$ (Equation (7)). The group match scores $\alpha_{gk}$ can be made sparse by using sparse normalizers like sparsemax or top-$k$ softmax, for example. Then, the relation between object $i$ and object $j$ only needs to be computed if there exists a soft group $G_k, k \in [n_g]$ which assigns non-zero weight to a discrete group $g \in \mathcal{G}$ which includes both $i$ and $j$. That is,

$$\mathcal{R} := \{R_{ij} \colon \exists\, k \in [n_g], g \in \mathcal{G} \text{ such that } \alpha_{gk} > 0 \text{ and } i, j \in g\}.$$

Thus, only a sparse subset of the $m \times m$ relation tensor would need to be computed and stored.

## C   GEOEMTRY OF REPRESENTATIONS LEARNED BY MD-IPR AND RELCONV

In this section, we explore and visualize the representations learned by MD-IPR and RelConv layers. In particular, we will visualize the representations produced by the RelConvNet model trained on the SET task described in Section 5.2. Recall that the MD-IPR layer learns encoders $\varphi_1, \psi_1, \ldots, \varphi_{d_r}, \psi_{d_r}$. In this model $d_r = 16$, $\varphi_i = \psi_i$ (so that learned relations are symmetric), and each $\varphi_i$ is a linear transformation to $d_{\mathrm{proj}} = 4$-dimensional space. The representations learned by a selection of 6 encoders is visualized in Figure 7. For each of the 81 possible SET cards, we apply each encoder in the MD-IPR layer, reduce to 2-dimensions via PCA, and visualize how each encoder separates the 4 attributes: number, color, fill, and shape. Observe, for example, that "Encoder 0" disentangles color and shape, "Encoder 2" disentangles fill, and "Encoder 3" disentangles number.

Next, we visualize, we explore the geometry of learned representations of relation vectors. That is, the inner products producing the 16-dimensional relation vector for each pair of objects. For each $\binom{81}{2}$ pairs of SET cards, we compute the 16-dimensional relation vector learned by the MD-IPR layer, reduce to 2 dimensions via PCA, and visualize how the learned relation disentangles the latent same/different relations among the four attributes. This is shown in Figure 8. We see some separation of the underlying same/different relations among the four attributes, even with only two dimensions out of 16.

Finally, we visualize the representations learned by the relational convolution layer. Recall that this layer learns a set of graphlet filters $\boldsymbol{f} \in \mathbb{R}^{s \times s \times d_r \times n_f}$ which form templates of relational patterns against which groups of objects are compared. In our experiments, the filter size is $s = 3$ and the number of filters is $n_f = 16$. Hence, for each group $g$ of 3 SET cards, the relational convolution layer produces a 16-dimensional vector, $\langle R[g], \boldsymbol{f} \rangle_{\mathrm{rel}} \in \mathbb{R}^{n_f}$, summarizing the relational structure of the group. Of the $\binom{81}{3}$ possible triplets of SET cards, we create a balanced sample of "sets" and "non-sets". We then compute $\langle R[g], \boldsymbol{f} \rangle_{\mathrm{rel}}$ and reduce to 2 dimensions via PCA. Figure 9 strikingly shows that the representations learned by the relational convolution layer very clearly separate triplets of cards which form a set from those that don't form a set.

## D   SOME INITIAL IDEAS ON HIGHER-ORDER RELATIONAL TASKS

As noted in the discussion, the tasks considered in this paper are solvable by modeling second-order relations at most. One of the main innovations of the relational convolutions architecture over existing relational architectures is its compositionality and ability to model higher-order relations. An important direction of future research is to test the architecture's ability to model hierarchical relations of increasingly higher order. Constructing such benchmarks is a non-trivial task which requires careful thought and consideration. This was outside the scope of this paper, but we provide an initial discussion here which may be useful for constructing such benchmarks in future work.

**Propositional logic.** Consider evaluating boolean logic formula such as,

$$x_1 \wedge ((x_2 \vee x_3) \wedge ((\neg x_3 \wedge x_4) \vee (x_5 \wedge x_6 \wedge x_7))).$$

Evaluating this logical expression (in this form) requires iteratively grouping objects and computing the relations between them. For instance, we begin by computing the relation within $g_1 = (x_3, x_4)$ and the relation within $g_2 = (x_5, x_6, x_7)$, then we compute the relation between the groups $g_1$ and $g_2$, etc. For a task which involves logical reasoning of this hierarchical form, one might imagine the grouping layers in RelConvNet learning the relevant groups and the relational convolution operation

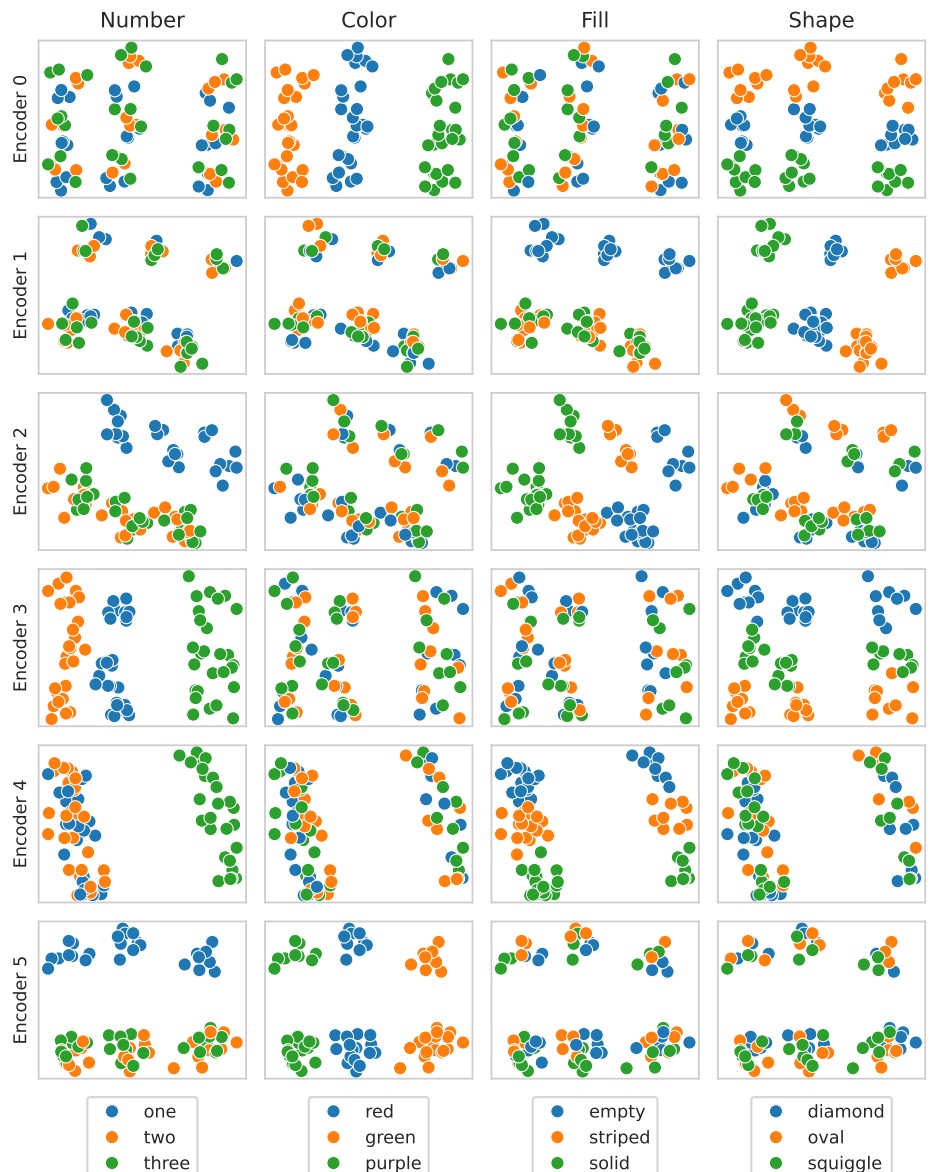

Figure 7: The encoders learned in the MD-IPR layer represent the latent attributes in the SET cards, with different encoders seemingly specializing to encode one or two attributes.

computing the relations within each group. Taking inspiration from logical reasoning with such hierarchical structure may lead to interesting benchmarks of higher-order relational representation.

**Sequence modeling.** In sequence modeling (e.g., language modeling), modeling the relations between objects is usually essential. For example, syntactic and semantic relations between words are crucial to parsing language. Higher-order relations are also important, capturing syntactic and semantic relational features across different locations in the text and across multiple length-scales and layers of hierarchy (see for example some relevant work in linguistics Frank et al., 2012; Rosario et al., 2002). The attention matrix in Transformers can be thought of as implicitly representing relations between tokens. It is possible that composing Transformer layers also learns hierarchical relations. However, as shown in this work and previous work on relational representation, Transformers have limited efficiency in representing relations. Thus, incorporating relational convolutions into Transformer-based sequence models may yield meaningful improvements in the relational aspects of sequence modeling. One way to do this is by cross-attending to a the sequence of relational objects produced

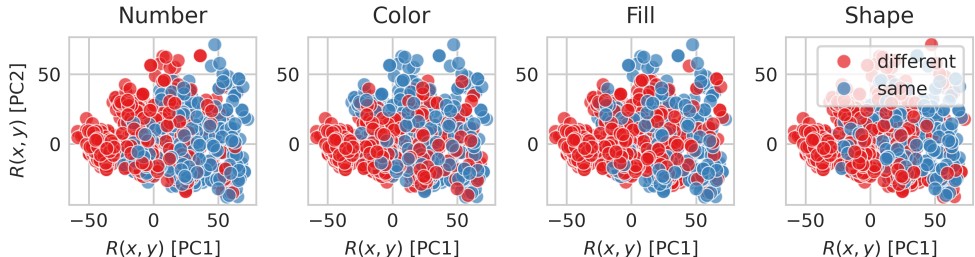

Figure 8: The relations learned by the MD-IPR layer encodes the latent relations underlying the SET task.

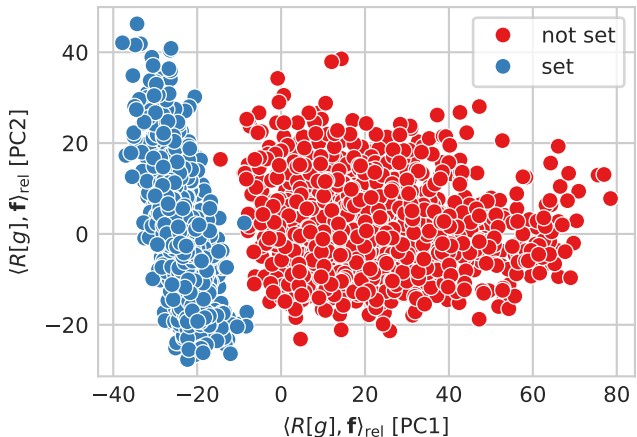

Figure 9: The relational convolution layer produces representations which separates 'sets' from 'non-sets'.

by relational convolutions, each of which summarizes the relations within a group of objects at some level of hierarchy.

**Set embedding.** The objective of set embedding is to map a collection of objects to a euclidean vector which represents the important features of the objects in the set (Zaheer et al., 2017). Depending on what the set embedding will be used for, it may need to represent a combination of object-level features and relational information, including perhaps relations of higher order. A set embedder which incorporates relational convolutions may be able to generate representations which summarize relations between objects at multiple layers of hierarchy.

**Visual scene understanding.** In a visual scene, there are typically several objects with spatial, visual, and semantic relations between them which are crucial for parsing the scene. The CLEVR benchmark on visual scene understanding (Johnson et al., 2017) was used in early work on relational representation (Santoro et al., 2017). In more complex situations, the objects in the scene may fall into natural groupings, and the spatial, visual, and semantic relations between those *groups* may be important for parsing a scene (e.g., objects forming larger components with functional dependence determined by the relations between them). Integrating relational convolutions into a visual scene understanding system may enable reasoning about such higher-order relations.

# E    DISCUSSION ON USE OF TCN IN EVALUATING RELATIONAL ARCHITECTURES

In Section 5.1 the CoRelNet model of Kerg et al. (2022) was among the baselines we compared to. In that work, the authors also evaluate their model on the relational games benchmark. A difference

between their experimental set up and ours is that they use a method called "context normalization" as a preprocessing step on the sequence of objects.

"Context normalization" was proposed by Webb et al. (2020). The proposal is simple: Given a sequence of objects, $(x_1, \ldots, x_m)$, and a set of context windows $\mathcal{W}_1, \ldots, \mathcal{W}_W \subset \{1, \ldots, m\}$ which partition the objects, each object is normalized along each dimension with respect to the other objects in its context. That is, $(z_1, \ldots, z_m) = \mathrm{CN}(x_1, \ldots, x_m)$ is computed as,

$$\mu_j^{(k)} = \frac{1}{|\mathcal{W}_k|} \sum_{t \in \mathcal{W}_k} (x_t)_j$$

$$\sigma_j^{(k)} = \sqrt{\frac{1}{|\mathcal{W}_k|} \sum_{t \in \mathcal{W}_k} \left( (x_t)_j - \mu_j^{(k)} \right)^2 + \varepsilon}$$

$$(z_t)_j = \gamma_j \left( \frac{(x_t)_j - \mu_j^{(k)}}{\sigma_j^{(k)}} \right) + \beta_j, \qquad \text{for } t \in \mathcal{W}_k$$

where $\gamma = (\gamma_1, \ldots, \gamma_d), \beta = (\beta_1, \ldots, \beta_d)$ are learnable gain and shift parameters for each dimension (initialized at 1 and 0, respectively, as with batch normalization). The context windows represent logical groupings of objects that are assumed to be known. For instance, (Kerg et al., 2022; Webb et al., 2021) consider a "relational match-to-sample" task where 3 pairs of objects are presented in sequence, and the task is to identify whether the relation in the first pair is the same as the relation in the second pair or the third pair. Here, the context windows would be the pairs of objects. In the relational games "match rows pattern" task, the context windows would be each row.

It is reported in (Kerg et al., 2022; Webb et al., 2021) that context normalization significantly accelerates learning and improves out-of-distribution generalization. Since (Kerg et al., 2022; Webb et al., 2021) use context normalization in their experiments, in this section we aim to explain our choice to exclude it. We argue that context normalization is a confounder and that an evaluation of relational architectures without such preprocessing is more informative.

To understand how context normalization works, consider first a context window of size 2, and let $\beta = 0, \gamma = 1$. Then, along each dimension, we have

$$\mathrm{CN}(x, x) = (0, 0),$$
$$\mathrm{CN}(x, y) = (\mathrm{sign}(x - y), \mathrm{sign}(y - x)).$$

In particular, what context normalization does when there are two objects is, along each dimension, output 0 if the value is the same, and $\pm 1$ if it is different (encoding whether it is larger or smaller). Hence, it makes the context-normalized output independent of the original feature representation. For tasks like relational games, where the key relation to model is same/different, this preprocessing is directly encoding this information in a "symbolic" way. In particular, for two objects $x_1, x_2$, context normalized to produce $z_1, z_2$, we have that $x_1 = x_2$ if and only if $\langle z_1, z_2 \rangle = 0$. This makes out-of-distribution generalization trivial, and does not properly test a relational architecture's ability to model the same/different relation.

Similarly, consider a context window of size 3. Then, along each dimension, we have,

$$\mathrm{CN}(x, x, x) = (0, 0, 0),$$
$$\mathrm{CN}(x, x, y) = \left( \frac{1}{\sqrt{2}} \mathrm{sign}(x - y), \frac{1}{\sqrt{2}} \mathrm{sign}(x - y), \frac{1}{\sqrt{2}} \mathrm{sign}(y - x) \right).$$

Again, context normalization symbolically encodes the relational pattern. For any triplet of objects, regardless of the values they take, context normalization produces identical output in the cases above. With context windows larger than 3, the behavior becomes more complex.

These properties of context normalization make it a confounder in the evaluation of relational architectures. In particular, for small context windows especially, context normalization symbolically encodes the relevant information. Experiments on relational architectures should evaluate the architectures' ability to *learn* those relations from data. Hence, we do not use context normalization in our experiments.

