# OpenReview forum: "Relational Convolutional Networks: A framework for learning representations of hierarchical relations"
_ICLR.cc/2024/Conference — Submitted to ICLR 2024_

### Official Review · Reviewer_Y5Cn · 2023-11-01

**Soundness:** 3 good
**Presentation:** 3 good
**Contribution:** 3 good
**Rating:** 6
**Confidence:** 2

**Summary:**

The paper proposes a framework for learning hierarchical relations analogous to the convolutional neural networks in the image domain. The authors propose a convolution operator with a graphlet filter to learn higher-level object relations. The method allows it to be directly applied to the learning problem without the need to explicitly model then which is the case in traditional Graph Learning. The method consists of an inner product relation module and relational convolution. The inner product relation module models pairwise relations between the input sequences. The relational convolution module uses the graphlet filter to extract features of relations between various object groups. The model is tested on relation games and set card datasets showing strong performance.

**Strengths:**

- The paper is well-written and easy to follow.
- The results show good performance as compared to the baseline on the set card dataset.
- The method allows us to learn higher-order relations between objects without explicitly modelling them.
- The proposed method is more sample-efficient than existing baselines.

**Weaknesses:**

The authors claim the method to be more interpretable and parameter-efficient manner, but there is no analysis for the same.

**Questions:**

Can the method be extended to be trained on knowledge graphs? This study can help us to understand the interpretability of the model easily due to the nature of the dataset as hierarchical relationships exist in knowledge graphs. It will be interesting to see how the method performs on it.

---

> ### Author Response · Authors · 2023-11-18
>
> Thank you very much for your feedback and your thoughts. We have incorporated feedback from you and other reviewers and made several additions to the paper. We believe this meaningfully improves the paper, and are grateful for this feedback. Please see the global response and the updated pdf for these additions; we hope you will take a look.
>
> ---
>
> > The authors claim the method to be more interpretable and parameter-efficient manner, but there is no analysis for the same.
>
> Thanks for the suggestion! We have added a section in the appendix analyzing the representations learned by the MD-IPR and RelConv layers on the "contains set" task. This is an appendix C of the updated pdf. The findings are quite interesting. We observe that the encoders in MD-IPR learn to encode and separate the four latent attributes in the task (number, color, fill, and shape). Moreover, the learned graphlet filters process these relations to *exactly separate* triplets of objects which form a "set" from those that do not. Please see Figure 7-9 in the appendix.
>
> We also added a discussion of parameter efficiency in Appendix B.1. The key idea is that the number of parameters in MD-IPR and RelConv is independent of the number of objects and the number of groups considered. This leads to computational and statistical benefits, as useful transformations are shared across pairs of objects and groups of objects. That is, analogous to the translation invariance of CNNs, the inner product relations are shared across pairs of objects and the graphlet filters are shared across groups of objects.
>
> Please take a look and let us know if this addresses your concern. We'd appreciate any further comments or questions you may have.

---

> > ### Comment · Reviewer_Y5Cn · 2023-11-22
> >
> > Thank you for the detailed response. I will maintain the current score

---

### Official Review · Reviewer_nJRP · 2023-11-01

**Soundness:** 3 good
**Presentation:** 3 good
**Contribution:** 3 good
**Rating:** 5
**Confidence:** 3

**Summary:**

The paper develops a novel convolutional framework named relational convolutional networks and demonstrate how relational convolutional networks can provide an effective framework for modeling relational tasks that have hierarchical structure.

**Strengths:**

1. A novel graph-liked CNN framework is developed for processing images
2. The developed method is techinical sounds and achieves promising performance on experimental evaluation

**Weaknesses:**

1. Some techinical details need to be claimed
2. More baselines should be considered for comparision

**Questions:**

Thanks for you awesome work. The authors develop a novel convolutional framework named relational convolutional networks to process images and demonstrate its effectiveness with a wide range of experiments.

1. More explanations about Fig.2 should be included, like symbol defination, and how you process pooling on graph.

2. In relation convolution layer, can you introduce the movitation of the design of attention-based pooling and more clear explanation about the techinical details?

3. In experiments, do you treat each pix of image as an token input for transformer? or a patch like ViT?

4. As your methods more like a GNN-based methods, I suppose more GNN-based baselines should be considered into comparsion, like Graph Transformer, Relation GNN etc.

---

> ### Author Response · Authors · 2023-11-18
>
> We appreciate your feedback. Please see the global response as well as the updated pdf for a description of some of the additions we made based on the feedback of reviewers. Below are some responses to your questions.
>
> ---
>
> > Some techinical details need to be claimed
>
> Can you clarify what you mean by this?
>
> ---
> > More explanations about Fig.2 should be included, like symbol defination, and how you process pooling on graph.
>
> Thanks for the suggestion. We have expanded the description in the caption. Please note that the notation is defined in the main body of the text, as well as in the annotations in the figure. The first row of the figure depict the graphlet filters which correspond to the parameters $\mathbf{f} \in \mathbb{R}^{s \times s \times d_r \times n_f}$. In this example, the filter size $s$ is 3. In the second row, the relation tensor $R$ is depicted as a graph with 5 nodes, and the red highlights correspond to each relation subtensor of $s=3$ objects---$g \in \mathcal{G}$ denotes a subset of $s$ objects, and $R[g]$ denotes the subtensor of $R$ corresponding to those objects (as defined in section 3.1 of the text). For each $g \in \mathcal{G}$, $R[g]$ is compared against the filter $\mathbf{f}$ via the relational inner product $\langle \cdot, \cdot \rangle_{\mathrm{rel}}$, as defined in equations 3 and 4. This yields the relational convolution $R \ast \mathbf{f}$ as defined in equation 6. Finally, given a group matrix $G \in \mathbb{R}^{m \times n_g}$, the relational convolution *given* $G$ is computed as described in equations 7-9.
>
> ---
>
> > In relation convolution layer, can you introduce the movitation of the design of attention-based pooling and more clear explanation about the techinical details?
>
> Are you asking about the context-aware grouping of equation 11?
>
> Here, the proposal is to generate a group matrix $G$ which assigns objects to soft groups using not only the object's own features, but also based on the context in which the objects appear. In general, this can be done via a message-passing neural network, as described in equation (11), by obtaining an embedding $E_i \gets \mathrm{MessagePassing}(x_i, \{x_1, \ldots, x_m\})$ for each object. The message-passing operation can learn to incorporate the relevant information about object $i$'s context in $E_i$ such that $E_i$ can be mapped to a vector $\mathrm{MLP}(E_i) \in \mathbb{R}^{n_g}$ encoding the object's group membership as a function of its context within the sequence. This contextual information may include information about the features of other objects in the sequence as well as their relation with object $i$.
>
> In the experiments, we use a Transformer Encoder block to encode the context (i.e., via self-attention). This is among the additions in the updated pdf. Please see section 5.1, Figure 5, and Table 3 in the appendix.
>
> ---
>
> > In experiments, do you treat each pix of image as an token input for transformer? or a patch like ViT?
>
> The input to the Transformer is a sequence of $m$ vector embeddings of each object image, obtained via a CNN. The sequence of objects are $(x_1, \ldots, x_m)$ where each $x_i \in \mathbb{R}^{w \times h \times 3}$ is an RGB image. Each image is processed independently using a CNN that produces a vector embedding $z_i \in \mathbb{R}^d$. The vector embeddings $(z_1, \ldots, z_m)$ are passed as input to the Transformer, just as for the other architectures. The CNN Embedder has a common architecture and the details are described in the appendix (Table 2 and Table 5, respectively, for each set of experiments).

---

### Official Review · Reviewer_ikDu · 2023-11-02

**Soundness:** 4 excellent
**Presentation:** 4 excellent
**Contribution:** 4 excellent
**Rating:** 8
**Confidence:** 4

**Summary:**

This paper tackles the problem of learning representations of relations between objects. Specifically, we are concerned with higher-order relations, that is, _relations between_ the relations between objects, and so on. This paper proposes a new neural network architecture by leveraging the principle of graphlet filters, to implicitly learn relations in an end-to-end fashion.

## Algorithm

1. We start with an initial set of $m$ objects, each represented by some vector.

1. **Multi-dimensional inner product:** These objects are embedded in some space by some non-linear operation, and multiplied with a linear embedding matrix. We take the dot product between the embeddings of each pair of objects, to get an $m \times m$ matrix representing the interactions between each pair. This process is repeated with $d_r$ different linear embedding matrices to get a $m \times m \times d_r$ tensor representing $d_r$ different kinds of interactions for each pair of objects.

1. **Relational Convolution:** Now that we’ve captured interactions between _pairs_, the next step is to represent interactions between _groups_. A graphlet filter is an $s \times s \times d_r$ tensor where $s \le m$. The $m$ objects are split into groups of $s$ objects each, and we perform an inner product between the $s \times s \times d_r$ graphlet filter and the $s \times s \times d_r$ pair-wise interaction sub-tensor from the larger $m \times m \times d_r$ tensor. This is meant to mimic image convolution with a filter.

    This will give us a single number. This process is repeated for $n_f$ different filters to get a vector of size $n_f$ representing different aspects of the group. If there are $|\mathcal{G}| = {m \choose s}$ different groups, then this gives us $|\mathcal{G}|$ different embeddings, each of size $n_f$, to carry to the next layer of the architecture.

4. **Grouping:** What if $|\mathcal{G}|$ is too large and we can only consider a maximum $n_g$ groups? The authors propose to learn $n_g$ “soft” groups, by using a learned “group matrix” $G$ of size $m \times n_g$. An entry of this matrix $(i, k)$ captures how much the $i$th element belongs to group $k$. Given some group of elements $g$, we can calculate their affinity for group $k$ by simply multiplying together the scores $G[i, k]$ of all the elements $i$ that belong to group $g$ (with some Softplus/Softmax stuff to take care of positivity and normalization). These affinity scores form a matrix of dimension $|\mathcal{G}| \times n_g$.

    This is where the embeddings of dimension $n_f$ from the previous layer come in. Each embedding corresponds to some group $g \in \mathcal{G}$. This forms a matrix of $n_f \times |\mathcal{G}|$. Multiply this with the affinity matrix $(|\mathcal{G}| \times n_g)$ to get $n_g$ embeddings, each of dimension $n_f$.

5. Through Steps 1, 2, and 3, we have transformed the representations of $m$ objects into $n_g$ representations, each corresponding to some grouping of the $m$ objects. This altogether comprises one layer of the RelConvNet, and there can be arbitrarily many layers to capture hierarchical relation information.

## Evaluation

The authors then compare their model with other models on two benchmarks - a “relational games” artificially constructed benchmark, and the game SET. They show that their model is able to learn complex relational tasks in a sample-efficient manner, and the gains are most pronounced for the most complex tasks.

Perhaps most impressively, the authors show that all the other models are _unable_ to learn the game of SET, at least with the train/test setting and the architecture that the authors considered.

**Strengths:**

1. Outstandingly well-written paper. Despite the technical intricacy, the paper has a great flow and is enjoyable to read; each section links to the next one in a very clear fashion, like a well-told story. It is honest about its limitations and precise about its strengths without exaggerating its claims.

1. The discussion about Transformers as message-passing networks with implicit relations was a new and interesting perspective for me.

1. The presented architecture is very very general, yet none of the design choices feel arbitrary. It is clear why the architecture achieves the inductive bias that the authors want. Its structure reflects its desired function.

1. The evaluation clearly shows the superiority of the proposed architecture for solving specialized relational reasoning problems. In particular, the result on SET was very impressive.

**Weaknesses:**

I have two main concerns with this paper:

1. Despite interpretability being presented as an advantage of this architecture, there is no qualitative analysis of the results on SET or relational games. It would be fun to see how the model has implicitly learned to group things according to shapes and patterns, and how the final decision comes together by aggregating all this information.

1. (This point is also mentioned by the authors in Limitations, but re-iterating). This paper’s approach is seemingly very powerful, capable of modeling interactions between many objects (with learned soft grouping) and learning higher-order relations beyond just second-order. But the evaluation is relatively tame in comparison, almost a toy example. Presented in this light, the paper seems almost like a solution in search of a problem, rather than the other way around. I feel like a short discussion about some real-world applications of such hierarchical reasoning would round-off the paper well.

Because of these two concerns, I gave this paper an overall score of 8 instead of the full 10.

**Questions:**

1. In Table 2 of the appendix, I notice that you’ve used only a single layer of MD-IPR and RelConv before flattening and MLP. If my understanding is correct, second-order relations need **two** layers to be represented accurately by your model. How is it that you’re still able to model SET and the “match pattern” accurately?

1. I can think of a much easier way to do the grouping layer described in Section 4 - just have a single linear transform of dimension $|\mathcal{G}| \times n_g$, with learnable parameters (and probably a Softmax). I assume you must have considered this and rejected it; could you give a short explanation of why? Is it because it is more parameter-efficient to use a matrix of size $m \times n_g$?

1. What are some real-world applications of higher-order relational reasoning that could make use of the full power of this model (with multiple layers and learned soft groups)? I know that one could construct artificial benchmarks, but I would like to qualitatively understand the problem too.

1. Could you provide some insight into the interpretability of what the model learns? For example, for the SET game, does one of the dimensions correspond to “colour”, one to “shape”, one to “number”?

---

> ### Author Response · Authors · 2023-11-18
>
> We are thrilled you found our paper interesting and enjoyable to read! Thank you for your thoughts, feedback, and suggestions. We have incorporated your feedback into an updated version of the paper. Please see the global response, updated pdf, and the response below for more details.
>
> ---
>
> > Despite interpretability...
>
> We agree that this would be interesting to see! We added a section to the appendix (section C) exploring and visualizing the representations learned by the MD-IPR and RelConv layers on the SET task. The results are interesting---we indeed see that different encoders in the MD-IPR layer learn to extract and compare different choices of the latent attributes in the data (i.e., number, color, fill, and shape). We also find that the RelConv layer learns graphlet filters which exactly separate triplets of cards which form 'sets' from those that don't! We hope you find this added analysis interesting.
>
> ---
>
> > (This point is also mentioned ...
>
> We entirely agree. The strength of this architectural framework is its ability to model hierarchical relations, but our experiments were limited to synthetic tasks relying on relations of first or second-order. While working on this project, we struggled to find benchmark tasks that fully test this ability. As you said, we found ourselves with "a solution in search of a problem," or more precisely, in search of a *benchmark*. We believe the ability to represent and reason about hierarchical relations has the potential to be useful in several interesting applications, including sequence modeling, set embedding, and visual scene understanding. We agree that the paper would benefit from more of a discussion on potential applications.
>
> We added an appendix section (section D) which discusses some ideas about applications of higher-order relational representation, and possible benchmarks that could be explored in future work.
>
> ---
>
> > In Table 2 of the appendix, ...
>
> Good question! Your intuition is correct that each additional layer of MD-IPR+RelConv models relations of one degree higher. In the case of the "match pattern'' task, the relations are indeed second-order, but the second-order relation is relatively simple. Having extracted the first-order relation among the triplet of objects, the MLP needs only to determine whether those first-order relations are the same or different. Thus, while a second layer of MD-IPR+RelConv would work here as well, it is not necessary.
>
> One way to think about this is that, given a sequence of objects, an MD-IPR layer computes first-order relations. Then, following a relational convolution layer, we obtain a sequence of objects, each summarizing the relational pattern within some group. Then, a second MD-IPR layer computes the relations between those relational objects, forming second-order relations. The RelConv layer in-between can be thought of as computing relations of an order that is part-way between first-order and second-order, because it has already grouped objects and computed the relational pattern between them.
>
> Because the final second-order relation is very simple (same or different), an MLP can parse the output of the RelConv layer into the final prediction.
>
> One noteworthy thing here is that, with the addition of learned grouping, RelConvNet can perform even better on the generalization split of the "match pattern" task in the relational games benchmark (see Figure 5 of the updated pdf where we have now added these results).
>
> ---
>
> > I can think of ...
>
> Parameter-efficiency is a big part of the reason. $\mathcal{G}$ will typically be exponential in the number of objects (e.g., of size $\binom{n}{s}$). This may be prohibitively large (computationally) when $n$ is moderately large. Another reason is the "statistical" aspect of how easy it would be to learn reasonable groups. We think this inductive bias makes learning meaningful groups easier because it links possible groups in $\mathcal{G}$, through object membership within them, rather than considering each group independently. In the case of temporal grouping, for example, group membership is determined by the temporal (positional) order in which the object occurs. It might be much more difficult to learn this kind of pattern with a parameterization in terms of $|\mathcal{G}|$ independent discrete groups instead. The $m \times n_g$ structure of the group matrix enables natural versions of the feature-based grouping and context-aware grouping layers as well, where an object's group assignment is simply a learned function of its features (i.e., $\phi(i, x_i) \in \mathbb{R}^{n_g}$ is a vector representing the degree to which the object belongs to each group).
>
> ---
>
> > What are some ...
>
> Please see Appendix D of the updated pdf :)
>
> ---
>
> > Could you provide some insight ...
>
> Yes! Please see Appendix C of the updated pdf.
>
> ---
>
> Please let us know if you have any thoughts about the additions or if you have any further questions or comments.

---

### Official Review · Reviewer_mJyy · 2023-11-04

**Soundness:** 3 good
**Presentation:** 3 good
**Contribution:** 2 fair
**Rating:** 5
**Confidence:** 4

**Summary:**

The paper proposes a new neural network architecture called "relational convolutional networks" for learning hierarchical relational representations.

The key components are:

* Multi-Dimensional Inner Product Relation (MD-IPR) module: Computes a multi-dimensional relation tensor between pairs of input objects using inner products. This aims to disentangle relational vs. non-relational features.

* Relational convolution layer: Convolves the relation tensor with a set of "graphlet filters", which represent templates of relations between subsets of objects. This extracts higher-order relational features.

* Grouping layers: Softly groups objects into relevant subsets over which to compute relations. This helps scale architecture.

* Overall architecture stacks MD-IPR and relational convolution layers to build hierarchical relational representations. This is analogous to CNNs building hierarchical spatial features.

**Strengths:**

* Relational convolutions provide an intuitive and interpretable way to build hierarchical relational representations, analogous to CNNs. The learned filters are inspectable.

* Experiments show the architecture is more sample efficient at relational reasoning tasks compared to Transformers and other baselines lacking explicit relational structure.

* The MD-IPR mechanism encourages disentangled relational vs. non-relational features through use of inner products. This is a simple but elegant inductive bias.

**Weaknesses:**

* Limited analysis of how the architecture scales as the number of objects and relations grow large. Memory and computation costs need investigation.

The MD-IPR module computes an $n \times n$ relation tensor for $n$ objects. This grows quadratically with $n$, so could become prohibitive for large $n$. The paper does not discuss optimizations like sparsity. In relational convolutions, naively considering all possible groups of objects scales exponentially. The grouping mechanisms help, but analysis of their scaling is lacking. Stacking many relational convolution layers could substantially grow the sequence length, increasing memory and computation per layer. The paper does not report metrics like parameter count or floating point operations. There is no experiment systematically increasing the number of objects and relations to analyze how performance degrades and costs increase. The experiments use fairly small input sizes.

* Main experiments are on simple synthetic tasks.

The relational games and SET tasks have at most simple second-order relations. Testing on tasks requiring modeling higher-order relations would better evaluate the capabilities of the relational convolutional networks. Besides, these tasks have a small fixed number of objects. Scaling to variable, larger numbers of objects would be more realistic.

* Lacks comparisons to recent related works on explicit relational reasoning.

The paper cites some prior works on relational reasoning like PrediNet and CoRelNet, but does not discuss or compare to other very recent methods such as Abstractors. By empirically comparing performance of relational convolutional networks to these contemporaneous methods on relational reasoning tasks, the unique advantages and tradeoffs of the proposed approach could be clarified. The lack of these head-to-head comparisons makes it harder to situate the advances of this architecture among other recent work. Adding these comparisons would significantly strengthen the paper.

**Questions:**

See the section of weakness above.

---

> ### Author Response · Authors · 2023-11-18
>
> We are grateful for your feedback. We have integrated it into an updated version of the paper, and believe that it has improved the final paper. Please see the global response, the updated pdf, and the responses below.
>
> ---
>
> > Limited analysis of how the architecture scales as the number of objects and relations grow large. Memory and computation costs need investigation.
>
> Thank you for the constructive suggestion! Indeed, it is possible to reduce the computation of the $n \times n$ relation tensor. Since the relational convolution considers relations *within a group*, we only need to compute relations between objects which co-occur in the same group. We have added an appendix section (B) which discusses how this can be done. Please also see the slightly-updated discussion on computing group-match scores in section 3.2. The key quantity is the set of discrete groups and group match scores, since a sparse relation tensor can be computed based on the "support" of those quantities. That is, $\mathcal{R} = \\{R_{ij} : \exists\, g \in \mathcal{G} \ \text{such that} \ i,j \in g\\}$ in the case of convolutions with fixed discrete groups, and $\mathcal{R} = \\{R_{ij} : \exists\, k \in [n_g], g \in \mathcal{G} \ \text{such that} \ \alpha_{gk} > 0 \ \text{and} \ i,j \in g\\}$ in the case of learned soft groups. By choosing a sparse normalizer in equation (7), $\alpha_{gk}$ will be sparse and hence $\mathcal{R}$ is sparse. For example sparsemax is used in the results added to the relational games experiments in section 5.1.
>
> ---
>
> > Main experiments are on simple synthetic tasks.
>
> We agree that more complex tasks, requiring modeling higher-order relations, would form an interesting evaluation of the architecture. Since existing relational architectures (e.g., PrediNet, CoRelNet, RelationalNet) are non-compositional, and did not focus on learning hierarchical relations, we were unable to find benchmarks for explicitly higher-order relational representation. The SET experiment is a step towards this, but we agree that evaluation on even higher-order relations as well as more realistic tasks would be interesting. The construction of such benchmarks requires careful thought and consideration, and was outside the scope of this project. However, we did add a section to the appendix (section D) to propose some early ideas on tasks involving higher-order relations and potential real-world applications of the relational convolutions architecture.
>
> ---
>
> A minor note:
> > Experiments show the architecture is more sample efficient at relational reasoning tasks compared to Transformers and other baselines lacking explicit relational structure.
>
> The PrediNet and CoRelNet architectures that we compare to are "explicitly relational architectures" from previous work on relational representation. You are likely aware of this, but we thought we'd clarify in case this point was missed. The difference is that those architectures lack the hierarchical/compositional structure of relational convolutional networks, making them less efficient on more difficult relational tasks (e.g., "match pattern" in the relational games), and completely unable to perform higher-order relational tasks (e.g., the SET experiments).
>
> ---
> Thanks again for your feedback and engagement with our work. We hope that this answers some of your questions and addresses some of your concerns. Please let us know if you have any other questions or feedback.

---

### Author Response · Authors · 2023-11-18
**Global response: additions to paper and integration of feedback**

Hi all. Thank you for taking the time to engage with our work and provide your thoughtful feedback. We appreciate your feedback and we believe it has improved our paper significantly. We look forward to discussing this work with you more.

We have updated our paper in light of your feedback and have uploaded the updated pdf. We would like to bring your attention to a few of the additions.
- In Section 5.1 (the experiments on relational games) we have added an evaluation of RelConvNet with learned grouping layers. In particular, Figure 5 now shows the out-of-distribution generalization performance of RelConvNet models with each of the three grouping layers proposed in section 4: temporal grouping, feature-based grouping, and contextual grouping. We observe that modeling groups explicitly results in performance improvements, particularly on the more difficult tasks and generalization splits.
- We added a section to the appendix (section B) discussing computational efficiency and parameter efficiency. In particular, we discuss how computing a dense $m \times m$ relation tensor can be avoided with a sparse set of discrete groups $\mathcal{G}$ and/or sparse group-match scores $\alpha_{gk}$ (see also the slightly expanded discussion on computing group match scores in section 3.2). We can instead compute a sparse set of relation vectors as a function of the support of the considered groups, $\mathcal{G}$ and $\alpha_{gk}$.
- We added a section to the appendix (section C) exploring and visualizing the representations learned by MD-IPR and RelConv layers. In particular, we observe that on the SET task, MD-IPR learns a relation function that separates the underlying latent relations over the four attributes, and the RelConv learns graphlet filters that exactly separate 'sets' from 'non-sets'.
- We added a section to the appendix (section D) which discusses some ideas about applications of higher-order relational representations, and possible benchmarks that could be explored in future work.

We hope that these additions address some of your questions and that you will take a look at these additions and share your thoughts. We look forward to further discussions.

---

### Meta-Review · Area_Chair_WAj2 · 2023-12-03

**Metareview:**

The paper proposes a new neural network architecture called "relational convolutional networks" for learning hierarchical relational representations. While the reviewers agree that this is an interesting direction, they also present salient arguments about the suitability of this paper for ICLR in its current form: limiting analysis of scaling behaviour, missing comparison to related work, main experiments on simple synthetic tasks only, and claims interpretability not quantitatively shown (e.g. using a user study).  We hope that the reviews are useful to you will push for one of the next DL venues.

**Justification For Why Not Higher Score:**

novel architecture that performes well for relational tasks. It even learns the abstractions.

**Justification For Why Not Lower Score:**

N/A

---

### Decision · Program_Chairs · 2024-01-16

Reject